# 3D Body Composition Segmentation in Abdomen and Pelvis CT using Subdivided Labels and Random Patch

**Minyoung Kim**[1,2]                                               KMY8381@EWHAIN.NET

[1] *Division of Mechanical and Biomedical Engineering, Ewha Womans University, Seoul, Korea*

[2] *Graduate Program in Smart Factory, Ewha Womans University, Seoul, Korea*

**Ji-Won Kwon**[3]                                                KWONJJANNG@YUHS.AC

[3] *Department of Orthopedic Surgery, Yonsei University College of Medicine, Seoul, Korea*

**Kwang Suk Lee**[4]                                              CALMENOW@YUHS.AC

[4] *Department of Urology, Yonsei University College of Medicine, Seoul, Korea*

**Taehoon Shin**[1,2]                                             TAEHOONS@EWHA.AC.KR

**Editors:** Under Review for MIDL 2023

## Abstract

The distribution and volume of fat and muscle in APCT play an important role as a biomarker. In this study, APCT data from 200 individuals who underwent health screening was labeled into three classes of fat and four classes of muscle. Based on this labeling, 3D patch-wise segmentation was performed by Swin UNETR on the whole abdomen and pelvic scan images. The test results showed an overall class average of 0.9227 DSC. This study conducted 3D whole-abdomen body composition segmentation using a total of eight segmented body composition labels including the background and verified its feasibility using random patches effective for the data and task.

**Keywords:** random patch, 3D, body composition, segmentation, APCT, Swin UNETR

## 1. Introduction

Information on body composition distribution is useful for diagnosing various diseases. Particularly, the analysis of the proportion and characteristics of fat and muscle is important for the diagnosis of diabetes and sarcopenia. Since manually assessing the distribution of body composition requires a significant amount of labor, numerous deep learning methods have been developed for automated segmentation. Previous body tissue segmentation models used 2D images or small-slab 3D volumes of abdominal-pelvic computer tomography (APCT) to reduce the computational load and labeling burden.(Koitka et al., 2021; Weston et al., 2019) In this study, we aimed to develop a 3D random patch-wise deep learning model for segmenting the entire 3D APCT into eight detailed types of fat and muscle tissues.

## 2. Materials and Methods

200 patients who visited the Gangnam Severance Hospital health promotion center between March 2021 and April 2021 and underwent APCT were selected. Segmentation classes were divided in detail into subcutaneous adipose tissue (SAT), visceral adipose tissue (VAT), external adipose tissue located outside the abdominal cavity (EAT), psoas muscles (PM), erector spinae muscles (ESM), multifidus muscles (MFM), remaining skeletal muscle classes

except for the above muscle classes (SM), and background, which belongs to none of the aforementioned classes. For annotation, the ITK Snap software(Yushkevich et al., 2006) was used and two orthopaedic and urologic surgeons with minimal 10 years of experience in the field participated in the labeling process. 180 and 20 subjects were used for training and testing, respectively. Data pre-processing included resampling to isotropic resolution of 1.5mm, background cropping, intensity normalization, and clamping between -300 and 500 in Hounsfield unit (HU).

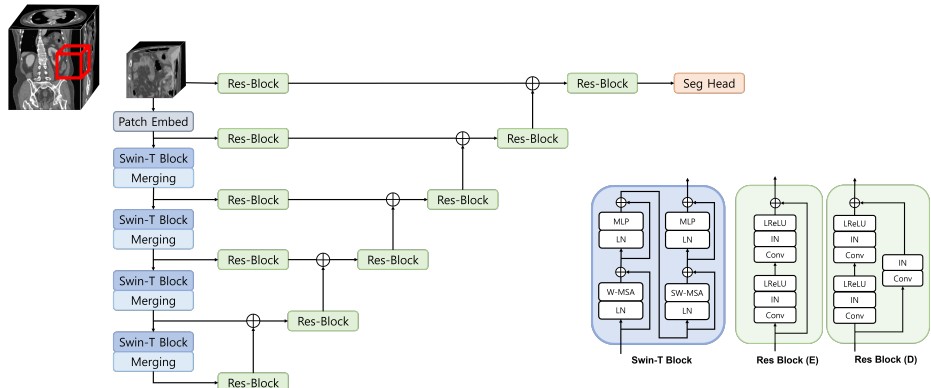

Figure 1: Overall Swin UNETR structure. Patch merging by 2 and the number of the heads are 3,6,12,24

Swin Unet Transformers (Swin UNETR)(Hatamizadeh et al., 2022) was used for 3D segmentation of the eight types of body tissues (including the background) (Fig. 1). Swin UNETR is a U-shaped model that combines an encoder with a Shifted window (Swin) Transformer architecture(Liu et al., 2021) and a decoder based on convolutional layers with skip connections. The Swin Transformer operates on self-attention within shifted windows of hierarchically varying resolutions, which decreases computational complexity and improves generalization performance. By employing a Swin transformer in the encoder, Swin UNETR can take advantage of hierarchical information during the upsampling process in the decoder. At each epoch of training, the model received randomly sampled 96x96x96 patches from 3D APCT images. This reduced the computational load and simplified the complexity of the whole abdominal pelvic image into patch units, allowing model to easily learn the features of each patch. Primary training parameters were batch size = 4, AdamW optimizer and Step scheduler with an initial learning rate of 0.001, step size of 400, and gamma of 0.5. Since the number of voxels substantially differs over the eight segmentation classes, we applied a hybrid loss that combines dice loss and cross entropy loss to address class imbalance(Taghanaki et al., 2019). For inference, we input grid-sampled patches from the test data into the model and combined the outputs to create a single segmentation for the entire trunk image. Accuracy was evaluated based on the Dice Similarity Coefficient (DSC).

## 3. Experiments and Results

Table 1 summarizes the DSC values obtained from the test dataset. On average across seven classes excluding the background, the patch size of 96 achieved the best performance

Table 1: Evaluation of Dice coefficient on the test set (P=patch size).

| Class | BG | SAT | SM | PM | VAT | EAT | ESM | MFM | Avg |
|---|---|---|---|---|---|---|---|---|---|
| **P=128** | 0.9874 | 0.9637 | 0.9323 | 0.9007 | 0.8685 | 0.7834 | 0.8788 | 0.8759 | 0.9014 |
| **P=96** | 0.9879 | 0.9635 | 0.9347 | 0.8988 | 0.8667 | 0.7815 | 0.8746 | 0.8689 | 0.9227 |
| **P=64** | 0.9868 | 0.9598 | 0.9339 | 0.8998 | 0.8399 | 0.7763 | 0.8691 | 0.8591 | 0.9180 |

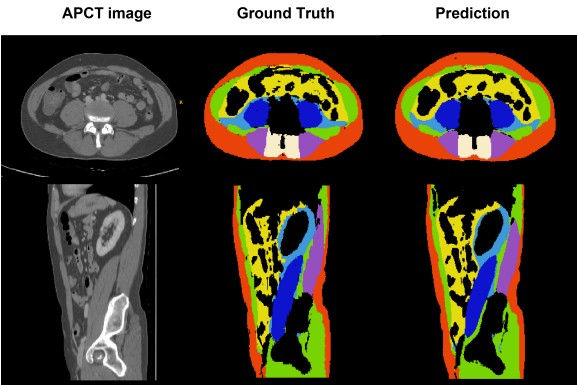

Figure 2: Axial and sagittal view images of the test subjects. SAT is indicated by red, VAT by yellow, EAT by light blue, SM by green, PM by blue, ESM by purple, and MFM by beige.

with average DSC of 0.9227. Figure 2 shows a representative set of raw APCT images, ground truth segmentation images, and segmentation images predicted from the model for the axial and sagittal views. VAT-EAT and MFM-ESM classes appear to be challenging due to adjacent tissues with the same HU values. The prediction results also show that the model has smoothed out uneven edges and discontinuities in the ground truth segments made during the labeling process.

## 4. Conclusion

We developed a deep learning models for automated segmentation of 3D whole abdomen into eight detailed fat and muscle regions. The proposed method used a Swin UNETR as a backbone with random patch-wise data augmentation. Test results showed an average DSC of 0.9227 over all eight classes, demonstrating the feasibility of the proposed approach.

## Acknowledgement

This research was supported by Digital Healthcare Research Grant through the Seokchun Caritas Foundation (No. SCY2208P), and the Institute of Information communications Technology Planning Evaluation (IITP) grant (No. RS-2022-00155966).

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
