# OpenReview forum: "3D Body Composition Segmentation in Abdomen and Pelvis CT using Subdivided Labels and Random Patch"
_MIDL.io/2023/Short_Paper_Track — MIDL 2023 Short paper track Poster_

### Official Review · Reviewer_Afy8 · 2023-04-15
**nice paper, strong results**

**Rating:** 8
**Confidence:** 4

**Review:**

nicely written paper, clear exposition

strong results

application has practical value in the clinic

---

### Official Review · Reviewer_pBjG · 2023-04-22

**Rating:** 5
**Confidence:** 5

**Review:**

Body composition estimation is useful for diabetes and sarcopenia diagnosis. To estimate the body composition, this paper proposes to segment abdomen and pelvis fat and muscle tissues using a patch-based V-Net. Experimental results demonstrate that the proposed method is able to segment the target tissues. However, there are two issues: (1) it is unclear how ground truth is obtained and how reliable the ground truth is; (2) the method V-net used in this study seems to be obsolete. The author should compare their patch-based V-Net with a more recent nn-Unet v2.